# Antimicrobial Activity and Action Mechanisms of Arg-Rich Short Analog Peptides Designed from the C-Terminal Loop Region of American Oyster Defensin (AOD)

**DOI:** 10.3390/md19080451

**Published:** 2021-08-06

**Authors:** Jung-Kil Seo, Dong-Gyun Kim, Ji-Eun Lee, Kwon-Sam Park, In-Ah Lee, Ki-Young Lee, Young-Ok Kim, Bo-Hye Nam

**Affiliations:** 1Department of Food Science and Biotechnology, Kunsan National University, Kunsan 54150, Korea; look77krok@naver.com (J.-E.L.); parkks@kunsan.ac.kr (K.-S.P.); 2Biotechnology Research Division, National Institute of Fisheries Science, Busan 46083, Korea; combikola@korea.kr (D.-G.K.); yobest12@korea.kr (Y.-O.K.); 3Department of Chemistry, Kunsan National University, Kunsan 54150, Korea; leeinah@kunsan.ac.kr; 4Department of Marine Biotechnology, Kunsan National University, Kunsan 54150, Korea; leekiy@kunsan.ac.kr

**Keywords:** antimicrobial peptide, American oyster defensin (AOD), Arg-rich analogs, antimicrobial mechanism

## Abstract

American oyster defensin (AOD) was previously purified from acidified gill extract of the American oyster, *Crassostrea virginica*. AOD is composed of 38 amino acids with three disulfide bonds and exhibits strong antimicrobial activity against Gram-positive bacteria as well as significant activity against Gram-negative bacteria. Here, to develop promising peptides into antibiotic candidates, we designed five arginine-rich analogs (A0, A1, A2, A3, and A4), predicted their loop and extended strand/random structures—including nine amino acids and a disulfide bond derived from the C-terminus of AOD—and described their antimicrobial and cytotoxic effects, as well as their modes of action. In our experimental results, the A3 and A4 analogs exhibited potent antimicrobial activity against all test organisms—including four Gram-positive bacteria, six Gram-negative bacteria, and *Candida albicans*—without cell toxicity. A sequence of experiments, including a membrane permeabilization assay, DNA binding study, and DNA polymerization inhibition test, indicated that the two analogs (A3 and A4) possibly did not act directly on the bacterial membrane but instead interacted with intracellular components such as DNA or DNA amplification reactions. AOD analogs also showed strong bacterial inhibition activity in the plasma environment. In addition, analog-treated microbial cells clearly exhibited membrane disruption, damage, and leakage of cytoplasmic contents. Collectively, our results suggest that two analogs, A3 and A4, have potent antimicrobial activity via DNA interaction and have the potential for development into novel antimicrobial agents.

## 1. Introduction

Over the past few decades, the number of antibiotic-resistant bacteria has increased due to the overuse of antibiotics and insufficient development of new antibiotics. These phenomena have necessitated the development of new antibiotics with different mechanisms of action than conventional antibiotics. In this regard, antimicrobial peptides (AMPs) are currently considered an attractive alternative, as they have different modes of action on the bacterial membrane compared to conventional antibiotics. AMPs have also been heralded as less harmful to both the environment and consumers, as well as capable of killing or inhibiting a pathogen without inducing resistance [1]. AMPs, which function as natural antibiotics in the animal kingdom, are important immune factors in the first line of defense in the innate immune system [2,3]. They are genetically encoded and produced by all types of living organisms, from prokaryotes to mammals. AMPs are considered promising drug candidates due to their broad range of activity, low toxicity, and decreased development of resistance by target cells [2,3].

American oyster defensin (AOD) was previously purified from the gill of the American oyster, *Crassostrea virginica*, and exhibits broad-spectrum antimicrobial activity without hemolysis [4]. This peptide contains three disulfide bridges linked with six cysteine (Cys) residues, which contribute to its structural stability. For the development of potential AMPs from AOD, the AOD molecule may be inappropriate due to its complex structure containing six Cys residues and long sequence length (38 residues), which complicate the recreation of its structure and increase production costs. In a previous study, Romestand et al. reported the antimicrobial activities of analog peptides derived from the C-terminus of the arthropod defensin *Mytilus galloprovincialis* defensin 1 (MGD-1), which was isolated from *Mytilus galloprovincialis* [5,6]. One promising analog peptide, consisting of the sequence CGGWHRLRC derived from the C-terminus of MGD-1, exhibited potent antimicrobial activity against several microorganisms [6]. In addition, the potent analog peptide CKKWKRKRC, derived from CGGWHRLRC by substitution of the residues at positions 2, 3, 5, and 7 with lysine (Lys), exhibited stronger antibacterial activity than the parent analog peptide [6]. These results suggest that MGD-1 analogs have potent antimicrobial activity against multiple bacterial strains, and that rearrangement of amino acids with net positive charge (Lys residues) is an important feature determining the strength of antimicrobial activity [6].

In several studies, tryptophan (Trp, W) and arginine (Arg, R) motif (WR motif) has been employed for analog peptide design, because tryptophan and arginine residues have strong membrane perturbation tendencies [7,8,9,10]. In addition, naturally occurring Arg-rich AMPs (containing WR motifs) in bivalves, Marine Mussels (*Mytilus* spp.), Myticalins, have been also reported [11]. Based on this property, we also selected and used the WR motif for analog design. Despite the similar physiochemical properties of Lys and Arg residues as the positively charged sources, their contributions to antimicrobial strength or other properties may differ [12]. In a previous study, an analogue containing the WR region exhibited stronger antimicrobial activity than an analogue containing the WK region [12]. These results indicate that the Arg residue is better for activity improvement in antimicrobial peptide design compared to the Lys residue [12]. According to these findings, we designed additional analogs through insertion of Arg residues into the sequence.

In this study, we designed five Arg-rich analogs for the improvement of antimicrobial activity, each consisting of nine amino acids with a disulfide bond derived from the C-terminus of AOD with high sequence homology to MGD-1, and investigated their biological activities against several pathogenic microorganisms as well as their modes of action to support the development of such peptides as antibiotic candidates.

## 2. Results

### 2.1. Subsection

#### 2.1.1. Analog Design and Synthesis

To select the template region for analog design, the tertiary structure of AOD (GenBank Accession No. EJ667947) was predicted using the homology modeling method (http://www.expasy.org/tools/) (Figure 1). The modeled structure showed that AOD is comprised of one α-helix (residues 9–17), two antiparallel β-strands (residues 21–25 and 33–36), and three loop regions (residues 1–8, 18–20, and 26–32) (Figure 1B, left). The modeled structure was similar to that of the arthropod defensin MGD-1, which has a Cys-stabilized α-β motif (Figure 1B, right). In a previous study, to investigate the structural motif important for antimicrobial function in MGD-1, Romestand et al. designed several analogs from the MGD-1 sequence and investigated their antimicrobial activities [6]. The region of loop 3 (CGGWHRLRC) in the C-terminus of MGD-1 was identified as a crucial region for antimicrobial activity. In general, loop regions of several AMPs and proteins have been proposed as crucial sites underlying antimicrobial or biological activities [6]. Thus, the C-terminal loop 3 region of AOD (CAGSLRLTC) was selected as the motif peptide for analog design, and the antimicrobial activity and modes of action of the resulting analogs were investigated. To design analogs from the selected region—loop 3 (CAGSLRLTC) in the C-terminus of AOD—physicochemical properties including primary structure, sequence length, pI value, net charge, number of Arg or Trp residues, and predicted secondary structure were first investigated (Table 1).

In this study, three strategies were used to design a series of AOD analogs: selection of a template region containing a disulfide bridge through homology modeling of AOD, substitution of amino acids to increase the net charge (or number of Arg residues) or alter polarity (through addition of Trp residues), and modification of the N- or the C-termini to increase activity. Table 1 shows the amino acid sequences and physiochemical properties of the AOD analogs used in this study. Collectively, the analogs were designed with high basic pI values (>10 except for A0), net positive charges (+1 to +6), and various numbers of Arg/Trp residues (1–6). The predicted secondary structures of the designed analogs were extended strand/random. Overall, five analogs (A0, A1, A2, A3, and A4) were designed and obtained through the solid-phase peptide synthesis method with good yields (>95%), which were dissolved in 0.01% HAc to obtain stock solutions of 1000 μg/mL.

#### 2.1.2. Antimicrobial Activity

To evaluate antimicrobial activity, MECs of the AOD analogs were determined against various strains of bacteria and the yeast *C. albicans* by URDA. The antimicrobial activities of analogs A2, A3, and A4 were potent, and were similar to or slightly weaker than the activity of piscidin 1, which was used as a positive control (Table 2). Among the AOD analogs, A2, A3, and A4 showed broader activity spectra than the template peptide A0, and exhibited potent antimicrobial activities against Gram-positive bacteria including *B. subtilis* and *S. mutans* (MECs of 0.4–10.5 μg/mL), as well as Gram-negative bacteria including *A. hydrophila*, two *E. coli* strains, *P. aeruginosa*, *S. sonnei*, and *S. enterica* (MECs of 2.0–29.5 μg/mL). By contrast, A1 had much weaker antimicrobial activity than the other analogs, and the parent peptide A0 exhibited no activity against any of the tested strains (Figure 2).

Notably, the most potent analogs, A3 and A4, also exhibited potent antimicrobial activity against *S. epidermidis* (MEC of 4.2–15.8 μg/mL), *C. acnes* (MEC of 53.7–27.1 μg/mL), and *C. albicans* (MEC of 54.0–17.2 μg/mL), which contribute to the development of acne (Table 2).

#### 2.1.3. Cytotoxicity of Analogs

To determine cytotoxicity, hemolytic activity of AOD analogs against human RBCs and their cytotoxic effect on HDFs were measured (Figure 3). None of the analogs caused significant hemolytic activity in human RBCs at concentrations from 3.13 to 100 μg/mL. By contrast, piscidin 1 caused strong hemolysis, even at very low concentrations (from 6.25 μg/mL) (Figure 3A). Furthermore, the cytotoxic effect of the AOD analogs on HDF cells was assessed through an MTT assay [13]. Cell viabilities following treatment with 50 and 100 μg/mL of each analog (A0, A1, A2, A3, and A4) were 88–89% and 87–88%, respectively (Figure 3B). These results indicate that AOD analogs have low cytotoxicity.

#### 2.1.4. Membrane Permeability

To determine whether the AOD analogs target bacterial membranes, the membrane permeability of the analogs was measured using *E. coli* ML35p for 60 min (Figure 4). Permeabilization of the *E. coli* ML35p inner cytoplasmic membrane was measured through the β-galactosidase assay [14]. *E. coli* ML35p was incubated with the AOD analogs or piscidin 1 and the chromogenic substrate ONPG, and the hydrolysis of ONPG by cytoplasmic β-galactosidase was verified spectrophotometrically at 405 nm. All AOD analogs exhibited similar basal levels of permeabilization or spontaneously penetrate bacterial membrane without leaking the intracellular components, while piscidin 1 led to strong permeabilization of the cytoplasmic membrane and leakage [15]. These results indicate that the mechanism of action of the AOD analogs may differ from that of piscidin 1, which directly acts on the bacterial membrane via the formation of toroidal pores, and that the target of the analogs may be related to intracellular components of the bacteria rather than the membrane [15].

#### 2.1.5. DNA Binding Assay and Investigation of the Weight Ratio of AOD Analog and DNA Needed for Interaction

To investigate whether the target site of the AOD analogs was an intracellular bacterial component, the DNA-binding ability of the AOD analogs was measured using an electrophoretic mobility shift assay (Figure 5A). The electrophoretic mobility of the DNA was inhibited by AOD analogs A1, A2, A3, and A4, and no DNA bands appeared in the inhibited lanes. The disappearance of these DNA bands might be due to the analogs blocking DNA binding by EtBr. AOD analogs bind to DNA preferentially over EtBr due to ionic interaction between the phosphate anion in the nucleotide and positively charged Arg residues in the analog [16]. These results suggest that AOD analogs may interact with intracellular DNA and thereby affect bacterial growth.

To determine the maximum binding ratio between AOD analog and DNA, DNA-binding ability was assessed based on the presence of a DNA band after mixing of AOD analog A3 and DNA in various weight ratios from 1:2.5 to 1:10 (DNA:A3) (Figure 5B). AOD analog A3 inhibited the migration of DNA at weight ratios greater than 1:2.5 (DNA:A3 = 0.1:0.25 μg). The migration of DNA was also inhibited at a weight ratio greater than 1:2.5 when a larger amount of DNA was used (DNA:A3 = 0.4:1.0 μg), and the band intensity in the well was sharply reduced. These results indicate that the interaction of analog A3 with DNA is direct and concentration-dependent, and that the loss of DNA band intensity in the well might be due to analog A3 blocking binding of EtBr to DNA [16].

#### 2.1.6. Competitive Binding Ability of AOD Analogs between DNA and DNA Polymerase

To assess competitive binding ability of AOD analogs between DNA and DNA polymerase during DNA processing, the preferential binding ability of AOD analog was assessed based on the presence of a DNA band in the gel after mixing of analogs with DNA and DNA polymerase followed by incubation at 37 °C for 60 min (Figure 6). The electrophoretic mobility of the DNA was inhibited by analogs A1, A2, A3, and A4, regardless of the presence of DNA polymerase. These findings indicate that the AOD analogs bind preferentially to DNA over DNA polymerase, whereas piscidin 1 preferentially binds to DNA polymerase over DNA, or piscidin 1 binding to DNA is blocked by DNA polymerase.

#### 2.1.7. DNA Polymerization Inhibition Assay

To determine whether the AOD analogs target DNA or DNA polymerase during the DNA polymerization reaction, the capacity of the AOD analogs to inhibit DNA polymerization was determined using *E. coli* gDNA as a template for PCR amplification (Figure 7A). Analog A2 did not inhibit the amplification of gDNA, while analogs A1, A3, and A4 partially or completely inhibited DNA amplification. Similar to analog A2, primer-only (P), 0.01% HAc, and piscidin 1 treatments, used as negative and positive controls, did not inhibit the amplification of gDNA. These results indicate that the target of analogs A1, A3, and A4 may be related to the inhibition of DNA amplification via direct binding to DNA, and that the binding ability of analog A2 to DNA may be inhibited during the DNA polymerization reaction.

To verify that the AOD analogs bind to other bacterial gDNA, the gDNA-binding ability of the most potent analogs, A3 and A4, was measured through an electrophoretic mobility shift assay using the gDNA of *Vibrio parahaemolyticus* (Figure 7B). The electrophoretic mobility of *V. parahaemolyticus* gDNA was inhibited by A3 and A4. These findings indicate that A3 and A4 can interact with bacterial intracellular gDNA and affect cell proliferation.

#### 2.1.8. Bacterial Inhibition Assay (BIA) in Plasma

To determine the stability and activity of AOD analogs under in vivo conditions, a BIA was performed with mouse plasma, which contains active components such as proteases (Figure 8). Incubation of bacteria with mouse plasma resulted in a slight reduction in the number of bacteria, but had a negligible impact on bacterial growth. Furthermore, analogs A0, A1, A2, A3, and A4 caused intensive growth inhibition of bacteria in plasma. These results indicate that the AOD analogs are stable and suitable for use under simulated plasma conditions and can be applied in vivo.

#### 2.1.9. Damage to Microbes Observed through Electron Microscopy

To confirm the effect of analog A4 on the morphology of microbes (*C. albicans*, *E. coli*, and *S. aureus*), SEM analysis was performed [17]. The control microbes had intact morphology, with round or rod-shaped surfaces showing no damage (Figure 9A,C,E). However, after treatment with analog A4, the morphology and surface shapes of all microbes were significantly altered, with shrinkage, visible damage, and visible debris, which were absent from the control (Figure 9B,D,F). These results indicate that microbes might be damaged or affected by analog A4 (Figure 9).

## 3. Discussion

The C-terminal loop region of AOD, purified from the American oyster *C. virginica*, was selected as the motif peptide for the design of AOD analogs [4]. In previous report, AOD is composed of 38 amino acid residues and showed broad antibacterial spectrum which has slightly stronger activity against Gram-positive bacteria than Gram-negative bacteria [4]. To increase antimicrobial activity and shorten the sequence length, three strategies were used to design a series of AOD analogs. First, to increase cationicity (Arg-richness) or adjust hydrophobicity, several residues (alanine, glycine, leucine, and threonine) in the selected region were substituted for a charged (Arg) or aromatic (Trp) residue, as indicated with underlining in Table 1. Trp was introduced to increase antimicrobial activity and reduce hemolytic activity [18]. Arg was also introduced to increase the net positive charge and thus increase ionic interaction with negatively charged phospholipids in bacterial membranes [8,19]. In several analog peptide designs or natural AMPs, Arg and Trp were employed or present in a “WR” motif due to their membrane perturbation tendencies [7,11,20]. Despite the very similar physiochemical properties of Lys and Arg in terms of cationicity, their contributions to antimicrobial properties differ. In a previous study, 8- and 10-mer WR peptides exhibited higher levels of antibacterial activity than analogs containing WK peptides [12]. For this reason, we used Arg residues to increase cationicity in the analog rather than Lys. To increase antimicrobial activity and stability, the N- or C-terminus can be modified to allow for acetylation or amidation [20,21]. In previous research, N-terminal acetylation of short Trp- and Arg-rich antimicrobial peptide analogs conferred notable resistance to serum aminopeptidases, while amidation at the C-terminus increased antimicrobial activity through introduction of a net positive charge [20]. Finally, a disulfide bridge was added to the terminal Cys residues in the peptides to improve their antimicrobial activity and serum stability [20]. According to these regards, five analogs (A0, A1, A2, A3, and A4) were designed and investigated their biological activity and their modes of action (Table 1).

Antimicrobial activity analysis of the AOD analogs indicated that the net charge (i.e., cationic strength based on the number of Arg residues) and number of Trp residues were more important to antimicrobial strength than terminal modifications or analog structures (Table 1 and Table 2). However, in previous study, it is mentioned that the configuration of AOD is presumably important for the interaction with the bacterial membrane even further studies are needed [4]. These results suggest that the mechanisms of action or effectors can be changed. Interestingly, A3, containing six Arg and one Trp residues, and A4, with four Arg and two Trp residues, showed strong and broad-spectrum antimicrobial activity against all of the tested strains—particularly acne-causing strains including *C. acnes*, *S. epidermidis*, and *C. albicans*—and no cytotoxicity (Figure 2 and Figure 3). These antimicrobial activities indicate that A3 and A4 are the most promising analogs for the development of antimicrobial agents (applied to the skin).

To determine the action mode of AOD analogs, membrane permeabilization ability via direct interaction with the membrane and DNA-binding ability via interactions with intracellular components were investigated. The membrane permeabilization study indicated that AOD analogs did not permeabilize the bacterial inner membrane or spontaneously penetrate membrane without leaking the intracellular components (Figure 4). In general, Arg-rich AMPs are able to spontaneously pass through the cell membrane without leaking or damaging cellular components [19]. Such cell-penetrating peptides (CPPs) are known as penetratins [22]. In addition, Arg residues endow peptides with positive charges and allow for ionic bonding necessary for interaction with the negatively charged phospholipids of the bacterial membrane [8]. Similarly, the AOD analogs also contain several Arg residues, which may also possibly contribute to spontaneous internalization of the membrane and pass through the cell membrane without leaking or damaging cellular components [19]. This result suggests that AOD analogs may possibly target intracellular components such as DNA rather than the bacterial membrane. Arg residues also endow peptides with positive charges and support the ionic bonding necessary for interaction with negatively charged nucleic acids. The AOD analogs contain several Arg residues, which may contribute to their membrane penetration and nucleic acid interaction capabilities [19,23]. However, the soft tick defensin, which has high sequence homology with AOD, is permeabilized the cytoplasmic membrane and caused lysis of Gram-positive bacteria [24]. This result suggests that AOD could be also possibly interacted and permeabilized the cytoplasmic membrane similar as the soft tick defensin even further studies are needed [4].

Studies on the interactions of AOD analogs with intracellular molecules (such as DNA) and their inhibition of DNA amplification indicate that AOD analogs directly bind to DNA in a concentration-dependent manner and preferentially bind to DNA over DNA polymerase (Figure 5 and Figure 6). However, the results of PCR amplification suggest that AOD analogs A1, A3, and A4 (presumably containing WRR motif) inhibit DNA amplification via directly/strongly binding to DNA, while the binding ability of AOD analog A2 (presumably containing WR motif) to DNA may be decreased by weaken ionic interaction or inhibited by some factors such as dNTPs or MgCl_2_ in the reaction solution, which requires further investigations (Figure 7A). Moreover, the AOD analogs can interact with the gDNA of two bacteria (Figure 7B). These results indicate that the target sites of AOD analogs may differ, with analogs A1, A3, and A4 targeting gDNA directly and A2 targeting gDNA-related reactions or processes, such as interference with nucleic acid synthesis, replication, or translation [25].

In addition, AOD analogs strongly inhibited bacterial growth in mouse plasma (Figure 8). These results suggest that compact conformation (due to a disulfide bridge between the terminal cysteines) and terminal modification of AOD analogs improve resistance to proteolytic degradation in the plasma environment, which contains diverse proteinases. In SEM analysis, AOD analog A4 was observed to significantly alter the morphology or surface structure of all microbes, causing shrinkage or other visible damage. These results indicate that the membrane penetrated AOD analog A4 can possibly affects intracellular metabolic reactions and induces cytoplasmic damage or leakage.

Collectively, all our experimental results suggest that two AOD analogs, A3 and A4, have potent antimicrobial activity through DNA interaction or inhibition of DNA process and will aid in the design of AMPs that target intracellular components or metabolic processes for development into novel antimicrobial agents.

## 4. Materials and Methods

### 4.1. Peptide Synthesis and Purification

Analogs (A0, A1, A2, A3, and A4) derived from the C-terminus of AOD were commercially synthesized by Peptron, Inc. (Daejeon, Korea) at a purity of >95%. Briefly, the peptides were synthesized using 9-fluorenylmethoxycarbonyl (Fmoc) solid-phase peptide synthesis with ASP48S (Peptron) and purified using reversed-phase high-performance liquid chromatography with a Vydac Everest C18 column (250 × 22 mm, 10 µm; Grace Davison Discovery Sciences, Deerfield, IL, USA). Elution was performed with a linear gradient of water/acetonitrile (3–40% (*v*/*v*) acetonitrile) containing 0.1% (*v*/*v*) trifluoroacetic acid. The molecular weights of the synthesized peptides were confirmed by liquid chromatography/mass spectrometry (HP1100 series; Agilent, Santa Clara, CA, USA). All synthetic peptides were dissolved in 0.01% acetic acid to obtain stock solutions of 1000 μg/mL.

### 4.2. Ultrasensitive Radial Diffusion Assay for Antimicrobial Potency

The antimicrobial activity of the AOD analogs was assessed with an ultrasensitive radial diffusion assay (URDA), as described previously [4]. The antimicrobial activity of the analogs (from 31.3 to 1000 μg/mL, 5 μL volume) was tested against several bacteria, including *Bacillus subtilis* KCTC1021, *Streptococcus mutans* KCCM40105, *Aeromonas hydrophila* KCTC2358, *Escherichia coli* D31, *E. coli* ML35p, *Pseudomonas aeruginosa* KCTC2004, *Salmonella enterica* KCTC2514, and *Shigella sonnei* KCTC2009. In addition, *Cutibacterium acnes* KCTC3314, *Staphylococcus epidermidis* KCTC1917, and *Candida albicans* KCTC7965 were used for the anti-acne activity test. All except *C. acnes* KCTC3314 were grown overnight for 18 h in tryptic soy broth (TSB; for bacteria) or Sabouraud dextrose broth (SDB; for *C. albicans*) at the appropriate temperature (25 °C for *A. hydrophila*, 37 °C for the others). *C. acnes* was grown for 72 h in reinforced clostridial medium (RCM) under anaerobic conditions with an anaerobic gas-generating pouch (GasPak EZ; Becton Dickinson and Company, Sparks, MD, USA) at 37 °C. Following overnight or 72-h incubation, the bacterial and *C. albicans* suspensions were diluted to a McFarland turbidity standard level of 0.5 (Vitek Colorimeter #52-1210; Hach, Loveland, CO, USA) corresponding to approximately 10^8^ colony forming units (CFU)/mL for bacteria and 10^6^ CFU/mL for *C. albicans*. Diluted bacterial or *C. albicans* suspension (0.5 mL) was added to 9.5 mL underlay gel containing 5 × 10^6^ or 5 × 10^4^ CFU/mL in 10 mM phosphate buffer (pH 6.6) with 0.03% TSB (or RCM) or 0.03% SDB and 1% Type I low-electroendosmosis agarose. The AOD analogs were serially diluted two-fold in 5 µL acidified water (0.01% acetic acid; HAc) and each dilution was added to 2.5-mm diameter wells in the 1-mm thick underlay gel. After incubation for 3 h at either 25 °C (for *A. hydrophila*) or 37 °C (for the others), the bacterial or yeast suspension was overlaid with 10 mL double-strength overlay gel containing 6% TSB, RCM, or SDB with 10 mM phosphate buffer (pH 6.6) in 1% agarose. Plates were incubated for an additional 18–24 h (72 h for *C. acnes*) and the clearing zone diameters were measured. After subtracting the diameter of the well (2.5 mm), the clearing zone diameter was expressed in units (0.1 mm = 1 U). 

The minimal effective concentration (MEC; μg/mL) of the AOD analogs was calculated as the X-intercept of a plot of clearing zone diameter units against the log_10_ of the peptide concentration [26]. Piscidin 1, an α-helical AMP isolated from hybrid striped bass (*Morone saxatilis* × *Morone chrysops*) was used as a positive control [27]. The antimicrobial assay was performed in triplicate and the results were averaged.

### 4.3. Hemolytic Activity Assay

The hemolytic activity of the AOD analogs was determined using human red blood cells (RBCs; blood type B) [26]. The RBCs were collected from heparin-treated blood through centrifugation at 3000× *g* for 5 min and washed three times with 50 mM phosphate buffer (pH 7.4) containing 150 mM NaCl to remove the plasma and buffy coat. A suspension of 3% hematocrit in buffer with or without peptides was incubated for 60 min at 37 °C. Hemolysis was expressed as the hemoglobin content obtained from the absorbance of the supernatant at 542 nm following centrifugation at 3000× *g* for 5 min, and the absorbance of 100% hemolysis was determined through measuring hemoglobin release after the addition of 0.1% Triton X-100. The peptide hemolysis percentage was calculated using the following formula:% Hemolysis = [(Abs_542nm_ in the peptide solution − Abs_542nm_ in buffer)/(Abs_542nm_ in 0.1% Triton X-100 − Abs_542nm_ in buffer)] × 100.

The hemolytic assay was performed in triplicate and the results were averaged.

### 4.4. Effect on Human Dermal Fibroblast Cell Viability

To assess the cytotoxic effects of AOD analogs on human dermal fibroblast (HDF) cells (Sigma-Aldrich, 106-05N), a 3-[4, 5-dimethylthiazol-2-yl]-2, 5 diphenyl tetrazolium bromide (MTT) assay was conducted [13]. Various concentrations of analogs (6.25, 12.5, 25, 50, and 100 μg/mL) were applied to HDF cells and cell viability was measured.

### 4.5. Membrane Permeabilization

To determine whether AOD analogs target the bacterial membrane, the extent of cytoplasmic membrane permeabilization was determined through measurement of β-galactosidase activity in *E. coli* ML35p using o-nitrophenyl-β-d-galactopyranoside (ONPG), a non-membrane-permeable chromogenic substrate [14]. Mid-logarithmic phase *E. coli* ML35p cells were washed in 10 mM sodium phosphate buffer (pH 7.4) and resuspended in the same buffer containing 1.5 mM ONPG. Hydrolysis of ONPG to o-nitrophenol over time was monitored spectrophotometrically at 405 nm following the addition of 40 μg/mL analogs. Membrane permeabilization assays were performed in triplicate and the results were averaged. Piscidin 1 and 0.01% HAc were used as positive and negative controls, respectively [27].

### 4.6. DNA Binding Assay

To evaluate the binding ability of the AOD analogs to intracellular components, a DNA binding assay was performed as previously described with minor modifications [28]. The DNA-binding ability of the AOD analogs was assessed through observation of the inhibition of migration of DNA bands through agarose gels. A commercial 100-bp DNA ladder (0.2 μg) (Bioneer Corp., Daejeon, South Korea) was mixed with each analog (1 μg) in 0.01% acetic acid, and the resulting mixture was incubated at 37 °C for 60 min and then electrophoresed on 1.8% agarose gels containing 0.5 μg/mL ethidium bromide (EtBr). HAc (0.01%) was used as a negative control.

### 4.7. DNA Binding Assay and Weight Ratio of AOD Analog and DNA Needed for Interaction

To determine the concentration required for the interaction of AOD analogs with DNA, a DNA binding assay was performed with various mixing ratios of analog A3 (0.25–1.0 μg/mL) and 100-bp DNA ladder (0.1–0.4 μg), as described above [16]. The concentration needed for the interaction of analog A3 with DNA was assessed based on the presence of a DNA band in the well that did not migrate through the agarose gel, as described above.

### 4.8. Competitive Binding of Analog between DNA and DNA Polymerase

To assess the preferential binding ability of the AOD analogs to DNA or DNA polymerase, a competitive binding assay was performed. Briefly, AOD analog was added to a reaction mixture containing DNA or DNA polymerase and incubated for 60 min at 37 °C. The reaction mixture was electrophoresed on 1.8% agarose gels and the preferential binding ability of the analogs was assessed through monitoring of a DNA band in the well that did not undergo migration. Piscidin 1 and 0.01% HAc were the positive and negative controls, respectively.

### 4.9. Genomic DNA Extraction and DNA Amplification Inhibition Assay

To determine whether AOD analogs target DNA amplification, the ability of the analogs to inhibit DNA amplification was assessed using *E. coli* DH5α genomic DNA (gDNA) [29]. Total gDNA was extracted using the phenol–chloroform method, as described previously [30]. To assess the inhibition of DNA amplification, 1.5 µg AOD analog was added to the polymerase chain reaction (PCR) mixture. The PCR solution consisted of 0.2 µg purified gDNA, 5 µL 10× PCR buffer with MgCl_2_, 4 µL of each dNTP at 2.5 mM, 0.25 μL Ex *Taq* polymerase (Takara Bio, Shiga, Japan), 2.0 pmol each of primers 27F (5′-AGAGTTTGATCCTGGCTCAG-3′) and 1492R (5′-TACGGCTACCTTGTTACGA-CTT-3′), and up to 50 µL sterile distilled water [31]. The PCR thermal cycling conditions were as follows: initial denaturation at 94 °C for 5 min; 30 cycles of 94 °C for 30 s, 55 °C for 30 s, and 72 °C for 2 min; and a final extension for 10 min at 72 °C. PCR was conducted using a Type 9700 Thermal Cycler (Perkin-Elmer, Norwalk City, CA, USA). The PCR products were confirmed through 1.5% agarose gel electrophoresis with 0.5 μg/mL EtBr. Piscidin 1 was used as a positive control and 0.01% HAc was used as a negative control.

### 4.10. Bacterial Inhibition Assay in Mouse Plasma

To determine the stability of the AOD analogs against peptidases, a bacterial inhibition assay (BIA) was performed using mouse plasma as previously described with minor modifications [32]. Mouse plasma was purchased from Sigma-Aldrich (St. Louis, MO, USA). Briefly, equal volumes of plasma (100 μL) and bacteria (*E. coli*, 1 × 10^3^ CFU/mL, 100 μL in soy broth) were incubated for 30 min at 37 °C. After the process was stopped by placing the mixture on ice for 15 min, AOD analogs (50 μg/mL, 100 μL) were added in a volume equal to that of the plasma/bacteria mixture and returned to the 37 °C shaker for 60 min. From the final mixture, 100 μL was smeared on solid medium and cultured at 37 °C for 24 h. The number of colonies was counted to determine bacterial abundance.

### 4.11. Scanning Electron Microscopy

The morphologies of *E. coli*, *S. aureus*, and *C. albicans* after treatment with analog A4 were observed using a scanning electron microscope (SEM) [17]. Mid-logarithmic phase cultures of *E. coli*, *S. aureus*, and *C. albicans* were prepared as described above and then resuspended in 495 μL fresh filtered phosphate-buffered saline. *E. coli*, *S. aureus*, and *C. albicans* were mixed with 10 μg of analog A4 for 2 h and immediately collected. The untreated cells were used as a control. The collected cells were subsequently fixed with 2.5% (*v*/*v*) glutaraldehyde in 0.1 M phosphate buffer (pH 7.4) overnight, washed three times with 0.1 M phosphate buffer (pH 7.4), and dehydrated using a graded ethanol series (30, 50, 70, 90, 95, and 100%). After critical point drying, the cells were mounted on 1-cm stubs and platinum-coated using a sputter coater (Q150T; Quorum Technologies, Lewes, UK). The specimens were then observed in SE2 mode (ETH = 5 kV) of the SEM (SUPRA-55VP; Carl Zeiss, Jena, Germany).

### 4.12. Structure Prediction and Homology Modeling

The secondary structures of AOD analogs were predicted through the Garnier–Osguthorpe–Robson (GOR) method (http://www.expasy.org/tools/) (accessed on 11 May 2021). The theoretical isoelectric point (pI) and net charge were estimated with the ExPASy ProtParam server (http://web.expasy.org/protparam/) (accessed on 17 May 2021). Sequence alignment was performed using Clustal X [33]. The structure of AOD was constructed through homology modeling using the SWISS-MODEL server (http://swissmodel.expasy.org/) (accessed on 16 March 2011). The structure of MGD-1 (Protein Data Bank Accession No. 1fjnA) was used as a template for modeling. The model structure was created with PyMOL (www.pymol.org) (accessed on 16 March 2011).

## 5. Conclusions

According to our investigations, AOD analogs A3 and A4 had potent broad-spectrum antimicrobial activity without cytotoxicity and could interact with intracellular molecules (such as direct DNA binding or effects on DNA amplification) to achieve antimicrobial functions. Therefore, AOD analogs A3 and A4 are promising candidates for the development of novel peptide antibiotics targeting bacterial DNA or DNA amplification.

## Figures and Tables

**Figure 1 marinedrugs-19-00451-f001:**
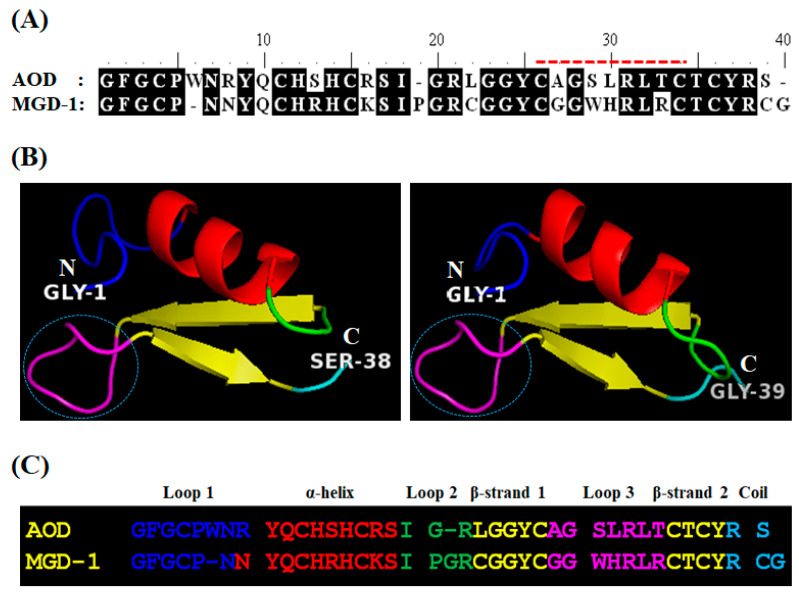
Homology modeling of American oyster defensin (AOD). (**A**) Amino acid sequence alignment of AOD with MGD-1. Conserved amino acids are indicated with black boxes. Amino acid residues indicated by the red dotted line were used for design of the analogs. (**B**) Homology modeling of AOD was performed using the crystal structure of MGD-1 (Protein Data Bank accession no. 1fjnA) as a template. Green, loops 1 and 2; red, α-helices 1 and 2; yellow, β-strands 1 and 2; cyan, C-terminal random region. The N- and C-termini of the peptides are indicated by N and C, respectively. The cyan dotted circle indicates loop 3, which was used for analog design. (**C**) Alignment of the AOD amino acid sequence with MGD-1. Colors are the same as in (**B**).

**Figure 2 marinedrugs-19-00451-f002:**
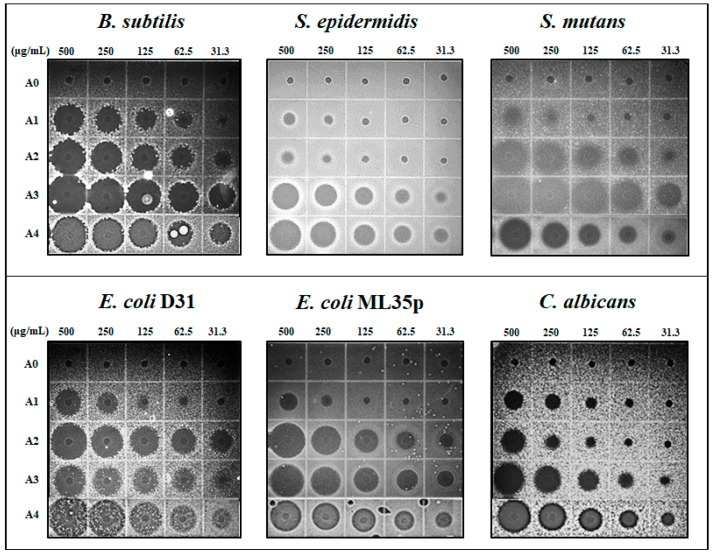
Antimicrobial activities of AOD analogs. Ultrasensitive radial diffusion assay of analogs (31.3–500 μg/mL, 5 μL volume) against *B. subtilis* KCTC1021, *S. epidermidis* KCTC1917, *S. mutans* KCCM40105, *E. coli* D31, *E. coli* ML35p, and *C. albicans *KCTC7965.

**Figure 3 marinedrugs-19-00451-f003:**
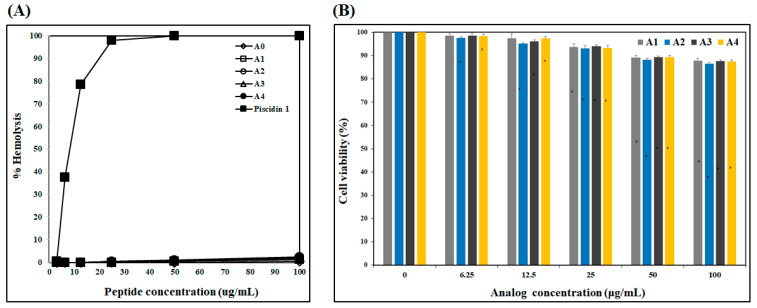
Cytotoxicity of AOD analogs and piscidin 1. (**A**) Hemolytic activity of the analogs and piscidin 1 against human erythrocytes (blood type B). Hemolysis was measured at analog concentrations of 3.13–100 μg/mL. (**B**) Cell viability of human dermal fibroblasts treated with analogs. Data are expressed as % of solvent control (Mean ± standard deviation, * *p* < 0.05, A1, 2, 3, 4 vs. solvent control).

**Figure 4 marinedrugs-19-00451-f004:**
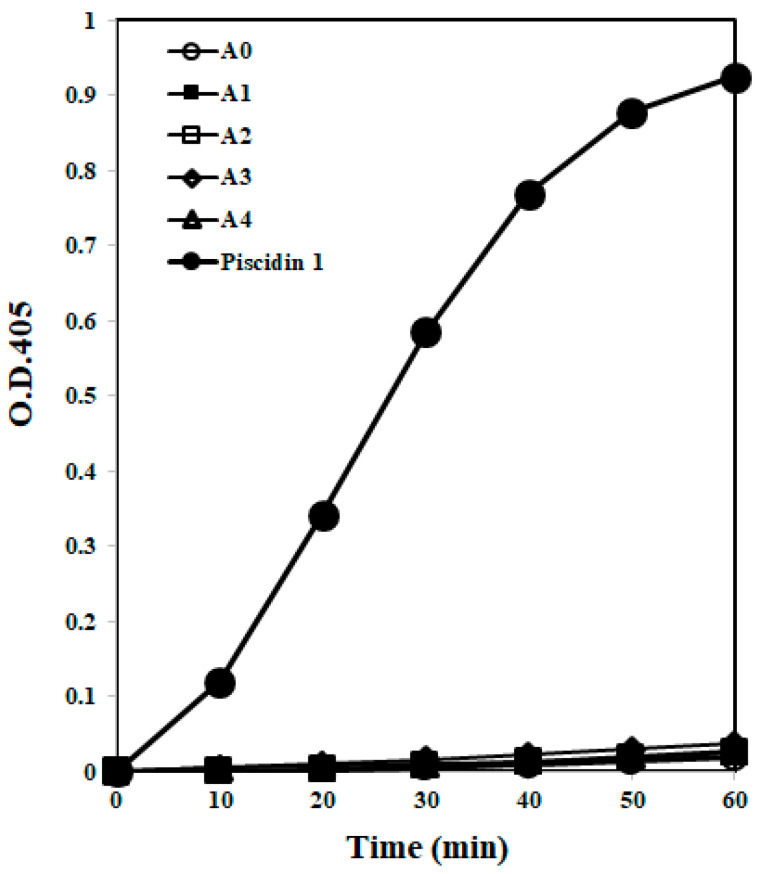
Cytoplasmic membrane permeabilization of *E. coli* ML35p by the AOD analogs and piscidin 1. Cytoplasmic membrane permeabilization was monitored at 405 nm, indicated by an increase in fluorescence intensity due to hydrolysis of the impermeable chromogenic substrate ONPG in the presence of analogs or piscidin 1 (40 μg/mL).

**Figure 5 marinedrugs-19-00451-f005:**
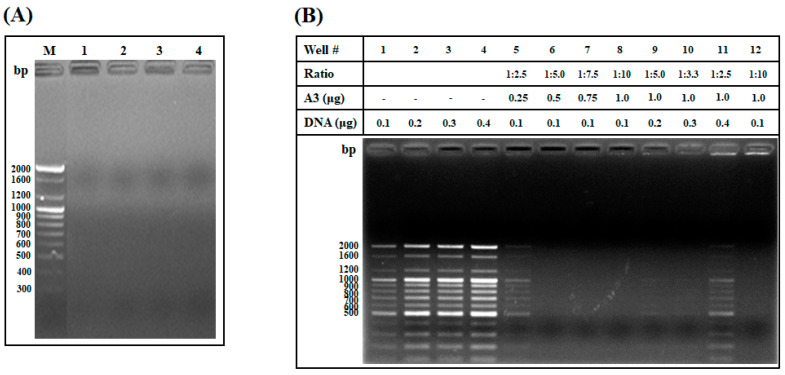
Gel retardation analysis of AOD analogs and weight ratios of AOD analog and DNA needed for interaction. (**A**) Binding of the AOD analogs (1 μg) to DNA was assessed by measuring the migration of a commercial 100-bp DNA ladder (0.2 μg) through an agarose gel. Each lane represents a mixture of 1 μg of A1, A2, A3, or A4 with 0.2 μg of 100-bp DNA ladder. M represents 0.2 μg of 100-bp DNA ladder with no peptide. (**B**) The concentration ratio that supported interaction of analog A3 with DNA was investigated using various weight ratios of analog A3 (0.25–1.0 μg/mL) to 100-bp DNA ladder (0.1–0.4 μg). Lanes 1–4 represent different amounts of 100-bp DNA ladder (0.1–0.4 μg) without analog A3. Lanes 5–8 represent mixtures of analog A3 (0.25–1.0 μg) with 100-bp DNA ladder (0.1 μg) (DNA:A3 = 1:2.5–1:10). Lanes 9–11 represent mixtures of 100-bp DNA ladder (0.2–0.4 μg) with analog A3 (1 μg) (DNA:A3 = 1:5.0–1:2.5). Lane 12 represents the mixture of 1 μg of piscidin 1 with 0.1 μg of 100-bp DNA ladder, used as a positive control.

**Figure 6 marinedrugs-19-00451-f006:**
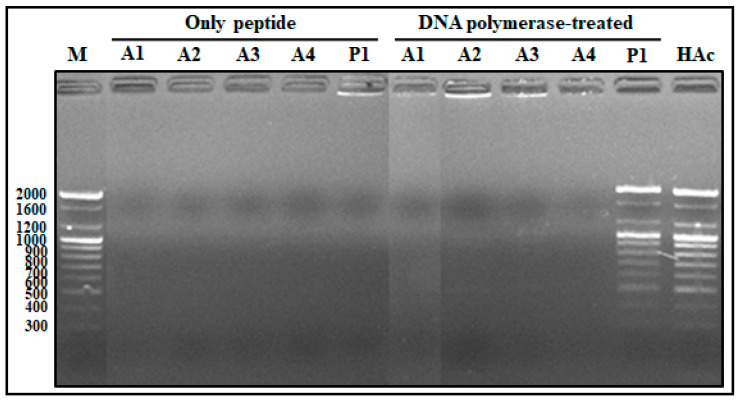
Competitive binding of AOD analog between DNA and DNA polymerase. AOD analogs (1 μg) were added to the reaction mixture containing 100-bp DNA ladder (0.2 μg) and DNA polymerase (1.0 μL, 5 units/μL) and incubated at 37 °C for 60 min. The reaction mixture was electrophoresed in 1.8% agarose gels and the preferential binding ability of the analogs was assessed through observation of a DNA band in the well. Only peptide: the mixture of each AOD analog (1 μg) with 100-bp DNA ladder (0.2 μg) without DNA polymerase; DNA-polymerase-treated: the mixture of AOD analog (1 μg) with DNA ladder (0.2 μg) and DNA polymerase (1.0 μL, 5 units/μL). Piscidin 1 and 0.01% HAc were used as the positive and negative controls, respectively.

**Figure 7 marinedrugs-19-00451-f007:**
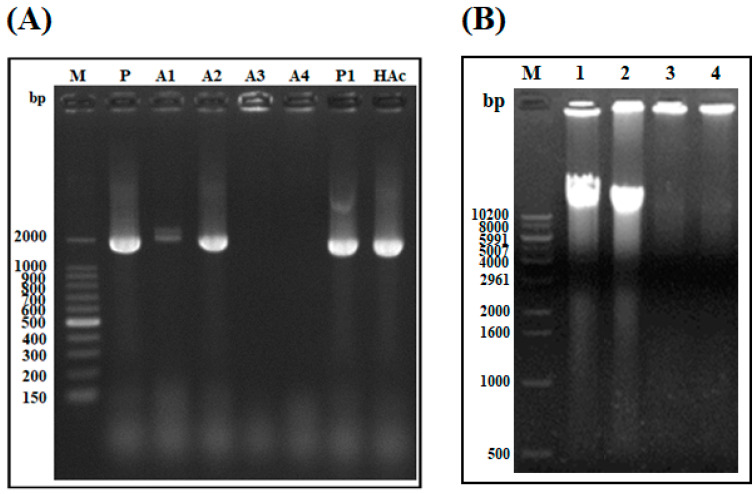
DNA amplification inhibition assay and bacterial gDNA-binding ability of the analogs. (**A**) The effects of the AOD analogs on DNA amplification were tested through PCR amplification of the 16S ribosomal gene from *E. coli* DH-5α gDNA in the presence of 1.5 µg of each analog. P, HAc, and P1 indicate primer-only, 0.01% acetic acid, and the positive control piscidin 1, respectively. (**B**) The binding abilities of the AOD analogs to bacterial gDNA were assessed through measurement of the migration of *V. parahaemolyticus* gDNA through an agarose gel. M: 0.2 μg of 1-kb DNA ladder; Lanes 1 and 2: *V. parahaemolyticus* gDNA and *V. parahaemolyticus* gDNA with 5 μL of 0.01% HAc, used as a negative control; Lanes 3 and 4: mixtures of 1 μg of analog A3 and A4, respectively, with *V. parahaemolyticus* gDNA.

**Figure 8 marinedrugs-19-00451-f008:**
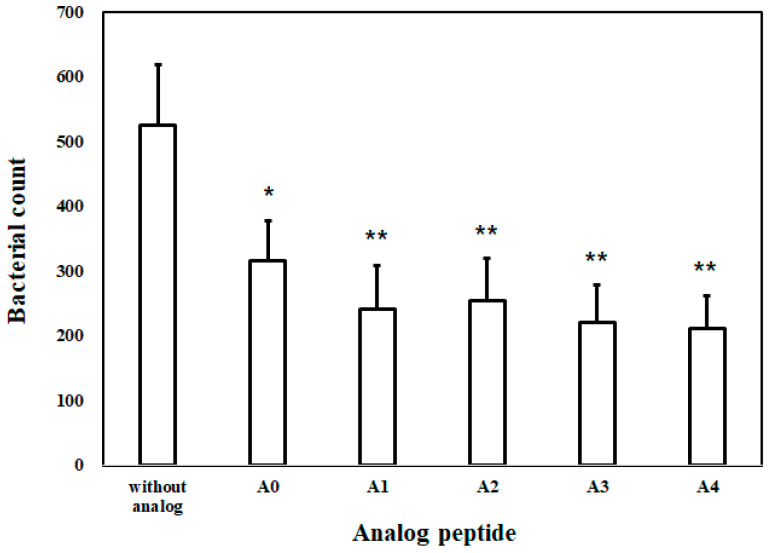
Bactericidal activities of the AOD analogs were assessed in mouse plasma using *E. coli*. Following incubation of the plasma/bacteria mixture for 30 min at 37 °C, the mixture was placed on ice for 15 min to stop bacterial activity, then the analogs (50 μg/mL, 100 μL) were added and the mixtures were incubated for 60 min at 37 °C. After incubation, 100 μL of the final mixture was smeared onto solid medium and cultured at 37 °C for 24 h. The number of colonies was counted to determine bacterial abundance. Asterisks (* or **) indicate significant differences between control and treatment (* *p* < 0.05, ** *p* < 0.01). Error bars indicate the standard deviation (n = 3).

**Figure 9 marinedrugs-19-00451-f009:**
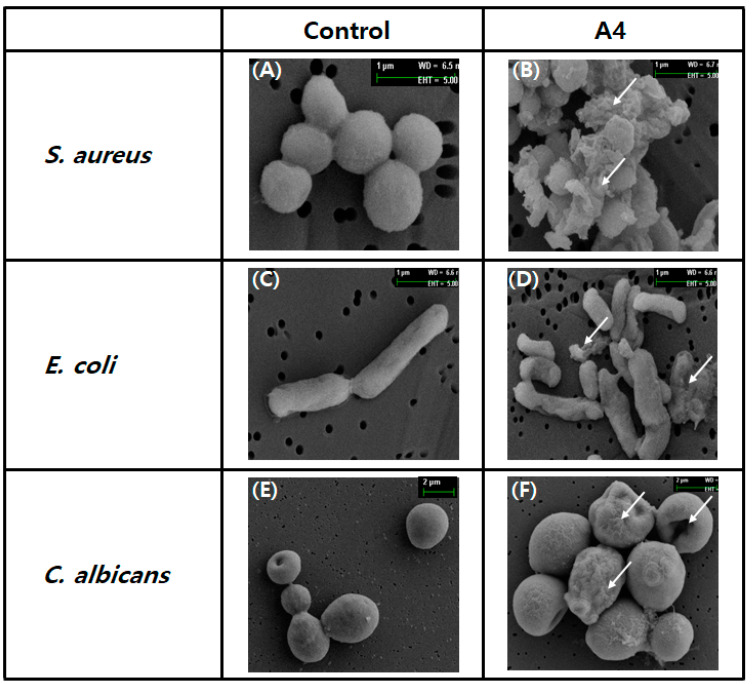
Morphological observations of microbes through scanning electron microscope analysis. *S. aureus* (**A**,**B**), *E. coli* (**C**,**D**), and *C. albicans* (**E**,**F**) were treated individually with 10 μg of AOD analog A4 for 2 h and then investigated. White arrows indicate coarse surfaces, cellular debris, or cell damage.

**Table 1 marinedrugs-19-00451-t001:** Physicochemical properties of AOD analogs.

Analog	Primary Structure	Length (aa)	pI ^a^	Net C ^b^	R ^c^	W ^d^	Predicted 2° Structure ^e^
A0	CAGSLRLTC-OH	9	8.07	+1	1	0	e and c
A1	Ac-CAGWRRLRC-NH_2_	9	10.4	+3	3	1	e and c
A2	CRWRLRLRC-OH	9	11.5	+4	4	1	e and c
A3	CRRWRRRRC-OH	9	12.0	+6	6	1	e and c
A4	CRRWGWRRC-NH_2_	9	11.53	+4	4	2	e and c

^a^ Analog isoelectric point based on ExPASy’s ProtParam server; ^b^ net charge (acetylation and amidation at the N- or C-terminus were not included) based on ExPASy’s ProtParam server; ^c^ number of Arg residues in each analog; ^d^ number of Trp residues in each analog; ^e^ secondary structure predicted with the GOR method (e: extended strand, c: random coil).

**Table 2 marinedrugs-19-00451-t002:** Antimicrobial activities of AOD analogs and piscidin 1.

Microbe	Gram Stain	Minimal Effective Concentration (μg/mL) ^a^					
		A0	A1	A2	A3	A4	Piscidin 1
*B. subtilis* KCTC1021	+	>100.0	14.0	3.8	0.4	2.5	2.3
*C. acnes* KCTC3314	+	>100.0	>100.0	>100.0	53.7	27.1	N.T. ^b^
*S. epidermidis* KCTC1917	+	>100.0	>100.0	24.0	4.2	15.8	N.T.
*S. mutans* KCCM40105	+	>100.0	68.0	9.9	1.5	10.5	3.2
*A. hydrophila* KCTC2358	-	>100.0	17.0	14.0	3.3	N.T.	9.1
*E. coli* D31	-	>100.0	42.0	2.0	5.0	5.7	2.0
*E. coli* ML35p	-	>100.0	>100.0	13.0	6.8	2.2	2.3
*P. aeruginosa* KCTC2004	-	>100.0	15.0	7.6	3.0	29.5	8.0
*S. enterica* KCTC2514	-	>100.0	24.0	20.0	6.5	N.T.	4.6
*S. sonnei* KCTC2009	-	>100.0	40.0	18.0	5.5	N.T.	8.6
*C. albicans* KCTC7965	Yeast	>125.0	>125.0	>125.0	54.0	17.2	11.8

^a^ Antimicrobial assays were performed in triplicate and the results were averaged; ^b^ N.T., not tested.

## Data Availability

Not applicable.

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
