# Peer review of "Antimicrobial Activity and Action Mechanisms of Arg-Rich Short Analog Peptides Designed from the C-Terminal Loop Region of American Oyster Defensin (AOD)"

_marinedrugs, 2021, doi:10.3390/md19080451_

Round 1

Reviewer 1 Report

The manuscript by Seo and colleagues report the creation and functional characterization of five arginine-rich analogs of the previously described American oyster defensing, a naturally produced antimicrobial peptide which plays an important role in the innate immune system of this filter-feeding bivalve.

This is an interesting and welcome study, that I feel was carried out in a competent way by the authors, using technically sound methods and a good rationale for peptide design. However, I think that the discussion might be improved to (i) provide more context about the known biological activity of the native unmodified peptide, which could help to better discuss the observed biological properties of the AOD analogs and discern whether they are likely to derive/be similar to those exerted by the peptide in nature; (ii) discuss more in depth the potential for a practical use, focusing in particular on the several aspects that are still unknown, such as about how the DNA interaction properties could explain the antibacterial action of these peptides in absence of peptide penetration inside bacterial cells.

Introduction: to further support the choice of including Arg residues instead of Lys, the authors could make reference to a few previously described naturally occurring Arg-rich AMPs in bivalves or other invertebrates. Myticalins are an example that comes to my mind (even though they are linear and carry no disulfide bond), but there may be several others, such as type III crustins in crustaceans.

Table 2 is affected by some visualization issues, with overlapping numbers. Please make sure to fix this.

Sections 2.1.4 and 2.15: how do the authors think the mode of action of these short AOD analogs may be related with the predicted mode of action of the full AOD (if this is known) or of the unmodified C-terminal region? Why was not the unmodified C-terminal region of the AOD peptide used as a control to check whether the small analogs displayed the same mode of action?

Also, could not the DNA-binding activity of the analogs be simply explained by the charge of the peptides themselves? In other words, would not the authors have expected to observe similar results for any other positively-charged peptide of a similar MW, regardless of its derivation from AOD?

Section 2.1.9. How do the authors explain these observations, considering the fact that in section 2.1.4 they indicate limited membrane interaction? This is very briefly discussed in lines 335-336, but should be in my opinion expanded.

Figure 1 legend: please cite the method used to perform homology modeling.

My feeling is that the discussion lacks a critical part concerning the specificity of the biological activities recorded by the authors for the AOD analogs. While they undoubtedly bind DNA, it is for example quite unclear whether this finding reflects a plausible biological activity carried out by such peptides in vivo (thinking about a practical application), or it is rather the by-product of their positive charge. In other words, in order to act on bacterial DNA, one might expect that such peptides are somehow able to penetrate bacterial cell membranes. I believe that adding some additional information concerning the know modes of action of other bivalve defensing-like AMPs may be useful to clarify this point.

Reviewer 2 Report

The manuscript entitled “Antimicrobial activity and action mechanisms of Arg-rich short analog peptides designed from the C-terminal loop region of American oyster defensin (AOD)” by Seo et al. introduces five new Arg-rich derived from the C-terminus of AOD with high sequence homology to MGD-1, and investigated their antimicrobial profile against several pathogenic microorganisms and inferred about their mode of action. Their goal was to introduce these peptides as potential antibiotic substitutes.

The work is very detailed and complete. Most importantly, the authors were able to offer a very simplified analysis of the results, which made it even easier of comprehension. However, they should alter Table 2 as it had a mistake. There are some English mistakes along the way as well, but overall, it was a very compelling and scientifically sound work. It would also be great if the discussion used more examples of the literature to highlight the novelty of this research and its impact in biomedicine. Regardless, the innovative nature of this work makes it very desirable for publication.
